# Immune Checkpoint Inhibitors’ Associated Renal Toxicity: A Series of 12 Cases

**DOI:** 10.3390/jcm11164786

**Published:** 2022-08-16

**Authors:** Kostas Palamaris, Dimitrios Alexandris, Kostas Stylianou, Ioannis Giatras, Anastasios Stofas, Christina Kaitatzoglou, Magda Migkou, Dimitrios Goutas, Erasmia Psimenou, Eleni Theodoropoulou, Stamatios Theocharis, Nektarios Alevizopoulos, Efstathios Kastritis, Alexandros Gerakis, Harikleia Gakiopoulou

**Affiliations:** 1First Department of Pathology, Medical School, National and Kapodistrian University of Athens, 11527 Athens, Greece; 2Department of Oncology, Evaggelismos General Hospital of Athens, 10676 Athens, Greece; 3Nephrology Department, Heraklion University Hospital, Voutes, 71500 Heraklion, Greece; 4Department of Nephrology, Hygeia Hospital, 15123 Marousi, Greece; 5Department of Clinical Therapeutics, School of Medicine, National and Kapodistrian University of Athens, 11528 Athens, Greece; 6Department of Nephrology, Tzaneio General Hospital of Pireaus, 18536 Pireas, Greece

**Keywords:** immunotherapy, immune checkpoint inhibitors, nephrotoxicity, tubulointerstitial nephritis, lupus-like membranous glomerulopathy, membranoproliferative glomerulonephritis, IgA glomerulonephritis, thrombotic microangiopathy, secondary AA amyloidosis, rheumatoid arthritis

## Abstract

We present a series of twelve patients, bearing a wide range of solid malignancies, who received either PD-L1 or a combination of PD-L1 and CTLA-4 inhibitors. Following immunotherapy administration, they exhibited the clinical signs indicative of renal toxicity, including increased serum creatinine levels, proteinuria, nephrotic syndrome and/or hematuria. All patients underwent renal biopsy. Results: All cases demonstrated some degree of interstitial inflammation and tubular injury, while in five patients, glomerular alterations consistent with a specific glomerulopathy were also observed: secondary “lupus-like” membranous glomerulopathy in two cases and membranoproliferative glomerulonephritis, IgA glomerulonephritis and secondary AA amyloidosis in each of the remaining three patients. The two patients with “lupus-like” nephritis and the one with amyloidosis experienced nephrotic syndrome, while their creatinine was within normal range. In the remaining nine cases, deterioration of renal function was the main manifestation. Conclusion: Our findings harmonize with bibliographical data that identify tubulointerstitial nephritis as the most frequent histological lesion related to ICIs administration. The preferential involvement of tubulointerstitial tissue could be associated with the reported higher expression levels of PD-L1 on tubular epithelial cells, compared to glomeruli. On the other hand, glomerular involvement is probably a consequence of a systemic immune system reconstruction, induced by immune-checkpoints inhibition.

## 1. Introduction

Immunotherapeutic approaches deploying immune checkpoint inhibitors (ICIs) have substantially prolonged patients’ survival in a multiplicity of solid and hematologic malignancies, taking advantage of their robust immunomodulatory activity. The most widely used immunotherapeutic modalities serve as inhibitors of PD-1/PD-L1 and CTL-4 checkpoint molecules. PD-1/PD-L1/PD-L2 and CTLA-4/CD80/CD86 axes act as intrinsic brakes that enable tumor cells to evade recognition by the immune system and hinder immune response [1]. They are also charged with a crucial role in preserving immune tolerance in homeostatic conditions and serve as crucial factors in containing inflammatory damage, during immune activation. Thus, it is a normal consequence of their physiological role in tissue homeostasis that their blockade incites a broad systematic reprogramming of host immune response by unleashing latent cellular sources of autoimmunity exacerbations [2]. They are therefore capable of evoking a unique spectrum of immune-mediated side effects with clinical manifestations and histological features that largely emulate autoimmune/inflammatory diseases involving the respective tissues. The multi-tissue nature of the ICIs-related adverse effects implies a possible combination of local and systemic factors as drivers of their pathogenesis [2]. Many organs can be affected simultaneously, with colon, skin and lung being among the most common targets. On the other hand, kidney is less frequently targeted by ICIs toxicity. The main manifestations of kidney involvement include renal failure, expressed as elevated serum creatinine levels, as well as the proteinuria of the subnephrotic or nephrotic range and hematuria, usually microscopic [2]. Renal biopsies conducted in patients with ICIs-induced renal toxicity have unraveled a variety of pathological entities that range from acute tubular necrosis and tubulointerstitial nephritis, occasionally with a granulomatous pattern [3,4,5,6], to a primarily glomerular involvement [7,8] in the form of podocytopahy (minimal change disease, focal segmental glomerulosclerosis, membranous and “lupus-like” glomerulopathy) [9,10,11,12,13,14,15] or as an inflammation-mediated glomerulonephritis (IgA glomerulonephritis, membranoproliferative glomerulonephritis, C3 glomerulonephritis and pauci-immune glomerulonephritis) [16,17,18,19,20,21,22].

In this paper, we present a series of twelve cases of ICIs-related kidney toxicity and summarize their clinical and histological features.

## 2. Materials and Methods

Our cohort includes twelve patients bearing various solid malignancies under immunotherapeutic regimens who experienced kidney toxicity either in the form of renal impairment with increased serum creatinine or as nephrotic syndrome. In order to identify the exact pathological background behind renal dysfunction, a renal biopsy was conducted in all the twelve patients.

All renal biopsies were processed for light microscopy (paraffin sections cut at 3 μm), immunofluorescence microscopy (cryostat sections for IgG, IgA, IgM, C3, C1q, C4, κ and λ light chains, albumin and fibrinogen) and—when tissue was sufficient—for electron microscopy. Histochemical stains (pas, Masson, silver, Congo-red) were performed in all cases and additional immunohistochemical stains (AA, p-component, ΙgG4, PLA2R etc.) when needed.

Interstitial inflammation, interstitial fibrosis and tubular atrophy (IF/TA) were evaluated semi-quantitatively in renal cortex as follows: mild (<25%), moderate (26–50%) and severe (>50% of renal cortex area). 

PD-L1 expression was estimated in all biopsies, using either anti-PD-L1 clone SP142 (Ventana Medical Systems, Inc., Tucson, AZ, USA) or (in patient 10) anti-PD-L1 clone 22C3 (PharmDx Dako Inc., Carpinteria, CA, USA). 

## 3. Results

### 3.1. Demographics-Clinical Characteristics

The median age of our patients was 66 years old (range: 57–73), eight of them were male, while the rest were female (male/female ratio: 2/1). The most common neoplasm was lung cancer, encountered in half of cases (n = 6), followed by bladder cancer in two patients (n = 2). Each of the remaining four patients had melanoma, renal cell carcinoma, squamous cell cervical carcinoma and Merkel cell carcinoma, respectively. The immunotherapeutic regimens administered encompass a variety of PD-1 (pembrolizumab, nivolumab) and PD-L1 (atezolizumab, avelumab) inhibitors, as well as anti-CTLA-4 agent ipilimumab. In addition to ICIs, four patients had also received proton pump inhibitors (PPIs). Within an average latent period of 14 months (range: 3–28) upon initiating ICIs treatment, patients experienced common manifestations of renal impairment, such as elevated creatinine levels or proteinuria, either of nephrotic or non-nephrotic range, including cases with full-blown nephrotic syndrome and hematuria. More specifically, nine patients demonstrated the deterioration of renal function with an average creatinine level of 2 mg/dL (range: 1.8–4.26 mg/dL). In three of them, renal failure was accompanied by hematuria, while four patients experienced proteinuria of non-nephrotic range with an average value of 24 h-urine protein of 1.14 g (range: 0.254–3.2 g/24 h urine). The three remaining patients experienced a full spectrum of nephrotic syndrome manifestations, including heavy proteinuria (10 g/24 h, 11 g/24 h and 16 g/24 h urine protein, respectively), hypoalbuminemia, dyslipidemia and peripheral edema. Interestingly, in one case, nephrotic syndrome followed the development of systemic autoimmune symptoms, compatible with rheumatoid arthritis. Besides renal involvement, five patients demonstrated additional adverse effects from other organs, such as liver, lung, colon and skin involvement, implying the multi-system nature of ICIs toxicity. Analytical clinical data regarding our patients are available in Table 1.

### 3.2. Histological Findings

All patients demonstrated some degree of interstitial inflammation, accompanied, in six cases, by acute tubular injury (ATI). Moreover, in five patients, glomerular lesions compatible with discrete glomerulopathies were also encountered: histological and immunofluorescent findings were consistent with lupus-like membranous nephropathy in two cases, and IgA glomerulonephritis, membranoproliferative glomerulonephritis along with thrombotic microangiopathy (TMA) and amyloidosis in each of the remaining three cases (Table 2). Two renal biopsies did not included glomeruli and so, the evaluation of glomerular involvement was not feasible. A full-blown nephrotic syndrome was the clinical manifestation in the two cases with “lupus-like” membranous nephropathy and in the one with amyloidosis. The remaining nine patients presented with the deterioration of renal function.

Tubulointerstitial nephritis (TIN): mild to severe interstitial inflammation, composed predominantly of lymphocytes, the majority of T-cell lineage, along with scattered macrophages and occasional eosinophils was detected (Figure 1). The distribution of inflammatory infiltrate was either diffuse or consisted of immune cell aggregates focally arranged within renal parenchyma. Moderate or severe inflammation was observed in five cases (patients 1–5), accompanied by tubulitis lesions, with the infiltration of tubular epithelium by T-cells. Immunotherapeutic schemes administered in these patients included PD-L1/PD-1 inhibitor monotherapy (pembrolizumab, nivolumab, atezolizumab) in four cases and a combination of pembrolizumab and CTLA-4 inhibitor ipilimumab in one patient (patient 4). Average period between treatment initiation and moderate/severe TIN emergence was 13 months (range: 3–24 months). Despite the robust immune reaction and the dense tubulointerstitial inflammatory infiltrate, no immunoglobulin or complement deposition was detected at the tubules’ basement membranes. Finally, the majority of patients displayed mild to moderate levels of chronicity, as determined by the percentage of interstitial fibrosis, tubular atrophy (IF/TA) and global glomerulosclerosis, while one (patient 2) demonstrated severe IF/TA (Table 3).

Secondary “lupus-like” membranous glomerulopathy: Two patients (patients 10,11) received PD-1 inhibitor monotherapy with pembrolizumab and nivolumab, for 23 and 10 months, respectively, and developed full scale nephrotic syndrome manifestations, including heavy proteinuria (16 g/24 h and 11 g/24 h), hypoalbuminemia and peripheral edema. No deterioration of renal function or urine sediment activity were detected (Table 2). The histological evaluation of renal biopsy revealed findings consistent with secondary membranous glomerulopathy with “lupus-like” features. Glomeruli demonstrated the mild increase of mesangial cellularity and matrix expansion, along with a broad thickening of capillary basements membranes, accompanied by the formation of subepithelial spikes. Immunohistochemistry revealed the global granular deposition of C4d along GBMs, and immunofluorescence exhibited a “full house” pattern with finely granular IgG, IgM, IgA, C3 and C1q deposits along glomerular capillary walls. The immunohistochemical evaluation of markers indicative of primary membranous nephropathy (IgG4 and PLA2R) was negative (Figure 2). Regarding chronic lesions, the percentages of glomerular sclerosis were 18.18% and 25%, respectively, while moderate tubular atrophy and interstitial fibrosis were observed in both cases (Table 3).

IgA glomerulonephritis: A 73-years old male (patient 1) with a medical history of hypertension received PD-L1 inhibitor atezolizumab and experienced kidney injury within twelve months upon treatment initiation. His serum creatinine level was 2 mg/dL, while his urine analysis revealed microscopic hematuria, characterized by 15–20 erythrocytes per high power field (HPF), (Table 2). Erythrocytes had dysmorphic morphology, implying glomerular origin. Interestingly, prior to immunotherapy, no glomerular hematuria was detected in patient’s examinations and his renal function was normal.

Histological evaluation demonstrated a simultaneous glomerular and tubulointerstitial involvement, while immunofluorescence showed strong IgA (3+) mesangial granular deposition, compatible with IgA nephropathy (Figure 3). Mild IgG and C3 deposits were also encountered. Evaluation for IgM globulin, C1q and C4 complement factors was negative. Under a light microscope, most glomeruli exhibited mild mesangial hypercellularity with the concurrent moderate expansion of mesangial matrix. Neither active lesions (endocapillary hypercellularity, crescents) nor segmental scleroses were encountered. According to Oxford’s classification, this IgA nephropathy was scored as M1E0S0T1-C0. A few glomeruli also exhibited ischemic alterations, which could be, at least partly, attributed to the history of hypertension. The accompanying tubulointerstitial nephritis was defined by moderate but focally dense chronic inflammatory infiltration. No immunoglobulin molecules or complement factors were identified in tubular basement membranes. The chronicity index of renal parenchyma was moderate, with a ~40% global glomerulosclerosis and moderate tubular atrophy/interstitial fibrosis (Table 3). This could be linked to the synergistic detrimental effect of three distinct pathogenetic factors: immune-complex mediated and ischemic glomerular injury and tubulointerstitial inflammation. Taking into account the absence of active glomerular lesions, the patient’s renal function acute deterioration was attributed to tubulointerstitial inflammation. Moreover, considering the recent onset of glomerular hematuria, glomerulonephritis was most probably linked to immunotherapy.

Membranoproliferative glomerulonephritis/thrombotic microangiopathy: A 57-year-old patient (patient 7), under atezolizumab treatment for 16 months, demonstrated renal impairment with a creatinine level of 2.2 mg/dL, accompanied by subnephrotic-range proteinuria (3.2 g/24 h). Urine sediment was inactive (Table 2). Renal biopsy revealed many of the trademark features defining membranoproliferative glomerulonephritis and thrombotic microangiopathy: glomeruli displayed the accentuation of their lobular architecture, the expansion of mesangial matrix, the double contours of glomerular basement membranes and segmental intracapillary hypercellularity. Moreover, some glomerular capillary lumens were obliterated by fibrin thrombi. Immunofluorescence demonstrated the deposition of the three immunoglobulin classes (IgG, IgA, IgM) and C3 complement factor, mostly in a linear discontinuous pattern along the glomerular basement membranes as well as and in the mesangium (Figure 4). Acute tubular injury was also recorded, while no chronicity lesions, such as global glomerulosclerosis, tubular atrophy or interstitial fibrosis, were detected (Table 3). 

AA Amyloidosis: A 70-year-old female patient (patient 12) developed rheumatoid arthritis within 28 months upon treatment initiation with pembrolizumab. Among the most prominent manifestations of her systemic autoimmune reaction was the emergence of nephrotic syndrome, characterized by urine protein levels of 10 g/24 h, hypoalbuminemia and edema. Serum creatinine was within normal range and urine sediment was inactive (Table 2). Kidney biopsy demonstrated the expansion of glomerular mesangial area and the occupation of the walls of many arterioles and interlobular arteries by acellular eosinophilic aggregates. These amorphous deposits stained positively for histochemical stain Congo-Red, showing apple-green birefringence under polarized light, consistent with amyloid deposits. They also demonstrated diffuse reactivity for amyloid A protein (AA), suggesting a secondary form of renal amyloidosis (Figure 5). Immunofluorescent evaluation for the detection of immunoglobulins (IgG, IgM, IgA), light chains (κ, λ) and complement factors was negative. Electron microscope revealed randomly disposed, non-branching 7 to 15 nm fibrils. In glomeruli, fibrils occupied mesangial areas, replacing the normal mesangial matrix and extending segmentally into some capillary walls. Finally, moderate global glomerulosclerosis and mild tubular atrophy/interstitial fibrosis were observed (Table 3).

### 3.3. PD-L1 Expression Pattern

Immunohistochemical evaluation revealed focal PD-L1 expression in eleven of our cases, while the lack of PD-L1 staining was only observed in the patient who received exclusively anti-CTLA-4 treatment (patient 8) (Table 4). In ten of the eleven positive cases (Patients 1–7,9,11,12), PD-L1 was detected in tubular epithelial cells, displaying a granular cytoplasmic pattern, ranging from mild to severe intensity (Figure 6). In three biopsies, which demonstrated either moderate or intense cytoplasmic staining, granular membranous PD-L1 of similar intensity was also encountered (Patients 4,6,7). Among the three patients with moderate or severe membranous PD-L1 staining, one had severe TIN (Patient 4), while in the other two only mild interstitial inflammation was observed. In two cases, tubular PD-L1 expression was accompanied by mild glomerular staining (Patients 11,12), while in one patient (Patient 10), PD-L1 expression was restricted solely in the glomerular compartment of renal parenchyma (Figure 6F). In the three further patients who showcased glomerular PD-L1 expression, specific glomerulopathies were identified: lupus-like nephropathy in two and secondary AA amyloidosis in the third.

### 3.4. Treatment-Follow Up

Data regarding the therapeutic approach of the patients’ immune-mediated adverse effects and their response to treatment were available in eight of our cohort cases (patients 2–6,9,10,12) (Table 5). In all the examined cases, the treatment plan was based on a combination of immunotherapy withdrawal and immunosuppression. Prednisolone monotherapy was the immunosuppressive scheme utilized in eight patients, while in the case of rheumatoid arthritis-induced AA amyloidosis (patient 12), it was combined with colchicine. Seven patients experienced renal response upon immunosuppressive treatment, either in the form of recovery of serum creatinine within normal values, in the cases of kidney injury, or as regression of heavy proteinuria in the patients with nephrotic syndrome. One patient, whose kidney biopsy had revealed diffuse interstitial inflammation, along with severe fibrosis, failed to respond to immunosuppression and experienced disease progression, with the further precipitous deterioration of renal function, ultimately resulting in hemodialysis (patient 2).

## 4. Discussion

Our cohort includes 12 patients with a wide range of both tubulointerstitial and glomerular pathological entities, which have been linked to ICIs implementation. The exact pathogenesis of these immune-mediated adverse effects has not been thoroughly studied. However, the decisive role of PD-L1 and CTLA-4 in the maintenance of immune tolerance and tissue homeostasis postulates a disruption of such homeostatic networks as the initial trigger for the induction of a progressively magnifying local or systemic autoimmune reaction. 

Consistent with previous studies, the most prominent lesion encountered in our cohort was tubulointerstitial nephritis (TIN), defined by various levels of inflammation, from scattered aggregates of immune cells to a diffuse or multifocal pattern of dense infiltrates [23,24]. Various levels of tubular atrophy/interstitial fibrosis were also encountered. The dominance of T-cells within the inflammatory infiltrate and the absence of immunoglobulin and complement deposits along the tubular basement membranes is consistent with a cellular-immunity dominated mechanism, as the driving force of this autoimmune attack against tubular epithelium. The immunohistochemical analysis of PD-L1 in our cohort revealed focal cytoplasmic or membranous staining in the tubular epithelium, reconciling with findings from previous studies of patients who developed TIN upon exposure to PD-L1/PD-1 inhibitors [25]. The single PD-L1-negative specimen originated from a patient who had not received anti-PD-L1/anti-PD-1 agents. Indeed, the upregulation of PD-L1 upon stimulation by inflammatory cytokines and mediators, such as IFN-β and IFN-γ, has been verified with in vitro experiments [26]. Therefore, a possible mechanistic explanation presumes the existence of cytotoxic T-cells with specificity for the antigens of tubular epithelial cells, which remain in a latent state as a result of the PD-L1/PD-1-inhibitory axis. Thus, the therapeutic blockade of this immune-suppressive path unleashes the cytotoxic potential of these CD8 lymphocytes, enabling them to attack and destroy tubular epithelium [26]. The fact that we could not identify a specific correlation between PD-L1 expression levels and inflammation severity confirms the highly complex and multi-factorial nature of immune system regulatory networks. As immunotherapy-related adverse effects represent autoimmune disorders, one cannot explain them in the context of a simplistic view of linear association between PD-L1 expression levels and inflammation severity. On the contrary, PD-L1 upregulation seems to be just one step of a multi-stage process, affected by all the heterogenous factors that determine immune system functional states, such as the patients’ age, gender, previous antigenic exposure of infectious or drug origin, etc. Moreover, the different checkpoint inhibitors, therapy duration and even the patients’ malignancies themselves could play a major role in determining the final outcome of immune system reprogramming induced upon this kind of treatment.

Moreover, four of our patients refer a previous PPIs administration, which is regarded as an additional risk factor for the development of tubulointerstitial kidney inflammation. This class of drugs is believed to incite the proliferation and partial activation of T-cells populations, with specificity for antigenic epitopes expressed on tubular epithelium [27,28]. Two main theories have emerged to explain their immunomodulatory activity: the first is based on the concept of molecular mimicry, suggesting that PPIs constituents share structural similarities with tubular epithelial antigens and, as a result, drug exposure leads to the activation of drug-specific lymphocytic clones, which can also recognize epitopes expressed on kidney tubules. The second possible mechanism assumes that the drug can serve as hapten, which modifies the structure of intrinsic antigens in such a way that the effector components of adaptive immunity do not recognize them as self-antigens anymore. The two possible mechanisms could very likely co-exist. It is therefore plausible that PPIs stimulate the mobilization of T-cells populations specific for tubular epithelial antigens, which are prevented from eradicating tubular cells by the inhibitory PD-L1/PD-1 axis [29]. As a result, a PD-L1 blockade mitigates the tolerance of pre-existing resting T-cell populations towards their targeted tubular antigens. 

Regarding glomerular lesions, in our cohort, we identified four distinct histological patterns: lupus-like membranous glomerulopathy, IgA glomerulonephritis and membranoproliferative glomerulonephritis, while one of our patients demonstrated secondary AA amyloidosis with a rheumatoid arthritis background. In the patients with lupus-like glomerulopathy and in the case of AA amyloidosis, glomerular PD-L1 staining was detected. While the observation of PD-L1 in glomerular compartment implies a local disturbance of peripheral tolerance, similar to that encountered in TIN, it seems that in immunotherapy-induced glomerulopathies, a systemic disruption of immune tolerance networks takes place [30,31]. As the two evolutionary conserved suppressive paths targeted by ICIs interfere with different steps of immune response, the broader reprogramming of immune system could induce the imbalance of humoral immunity, leading to autoantibodies generation [30]. Indeed, many autoantibodies have been isolated in the serum of patients receiving ICIs, which could contribute to the formation of immune complexes that deposit on glomeruli. These immune complex aggregates elicit an inflammatory reaction, leading to a glomerulonephritis type of injury that emulates well-established glomerular lesions. Lupus nephritis is the most thoroughly studied glomerulopathy, and divergent evidence from mice-based experimental studies have identified the impairment of PD-L1/PD-1 axis immunosuppressive function as a critical step in disease pathogenesis [31]. The genetic deletion of PD-L1 in a mouse model leads to a multi-system autoimmune syndrome that involved various tissues, including the kidney. Renal lesions were characterized by a glomerulonephritis pattern compatible with lupus-like glomerulopathy [32]. Further experiments conducted on the most commonly deployed lupus murine model, consisting of a NZB × WF1 mouse strain, confirmed the protective role of the PD-L1/PD-1 axis in attenuating lupus-induced glomerulopathy. NZBxWF1 mice demonstrate a clinical and histological profile that closely resembles the respective human disease manifestations, as they develop proteinuria and lupus serological immune profile (ANA, ds-DNA etc.) associated with pathological features of lupus nephritis [33]. The administration of a PD-L1 agonist antibody in such mice alleviated proteinuria and significantly prolonged their survival. This phenotypic alteration was accompanied by diminished serum anti-dsDNA and markedly decreased IgG glomerular deposition [33].

Concerning the cases of IgA glomerulonephritis and membranoproliferative glomerulonephritis, no specific mechanistic studies attempting to elucidate the possible participation of the PD-L1/PD-1 axis on diseases pathophysiology have been conducted so far. However, as their pathogenesis is believed to be associated with excessive immune response to a variety of etiologic factors, a deregulation of immune tolerance mechanisms based on inhibitory receptors could elicit a similar exacerbation of immune reaction. In this context, a study of patients with IgA and membranoproliferative glomerulonephritis points towards a role of PD-L1/PD-1 in their pathophysiology. Increased populations of PD-1+ and PD-L1+ T-cells and B-cells were identified in patients’ serum, while their frequencies showed strong positive correlation with IgM levels [34].

Finally, the patient with secondary amyloidosis, developed in the setting of pembrolizumab-induced rheumatoid arthritis, represents the first case of AA amyloidosis where the initial autoimmune inflammatory trigger for amyloid deposition has been identified. A few more case reports, describing AA amyloidosis upon immunotherapy administration, failed to associate it with a specific inflammatory/autoimmune background [17,35]. 

Regarding the therapeutic protocol utilized in our patients, it was based in a combination of immunotherapy discontinuation and corticosteroids administration. In the case of rheumatoid arthritis, corticosteroids were accompanied by colchicine. The majority of our patients responded to immunosuppression treatment, except one, who required hemodialysis. In that specific case, a plausible explanation could be the higher levels of chronicity index in the form of tubular atrophy and interstitial fibrosis encountered in the patient’s biopsy, suggesting the extensive, irreversible damage of renal tissue, which could not be overturned by immunosuppression. A correlation of inflammation severity with patient response, as described in previous reports, was not observed in our study [36]. In general, steroids are the well-established first-line treatment for kidney immune-related adverse effects, and they are usually successful at inducing an initial regression of renal manifestations. Their efficacy was confirmed in our cohort, as renal response was observed in seven out of the eight patients. In the case of steroid-resistant side effects, additional agents deployed, including both traditional immunosuppressive drugs, such as mycophenolate-mofetil (MMF), and more recently developed immunomodulator agents, such as anti-CD20 (rituximab) and anti-TNF (infliximab) factors. Particularly infliximab has shown significant therapeutic potential in both interstitial inflammation and in cases of glomerulonephritis (IgA nephropathy) and seems to be a very promising second-line treatment option [17].

## 5. Conclusions

In this case series, we present a wide range of kidney lesions associated with ICIs. Although TIN is frequently reported, the different types of glomerular involvement have not been thoroughly studied in terms of morphologic and immunophenotypical characteristics. It is important for the oncologist and the nephrologist to consider that glomerular hematuria or nephrotic syndrome might be related to ICIs. Moreover, renal biopsy findings can mimic those of well-established glomerulopathies and so ICIs’ toxicity should be in the pathologic differential diagnosis. Notably, in our series, secondary AA amyloidosis in a pembrolizumab-induced rheumatoid arthritis background is reported for the first time. Moreover, this is, to our knowledge, the first time that PD-L1 expression has been observed in the glomeruli of patients with ICIs-induced glomerulopathies, offering a new dimension to their possible pathogenetic mechanisms. As the multi-systemic adverse effects impose severe restrictions to the effective implementation of ICIs, there is an utmost need for the best possible identification of the many different ways their toxicity is expressed in multiple tissues. Thus, the better characterization of both the clinical profile and the histological “footprints” of immunotherapy adverse effects will enable a more effective monitoring of patients, aiming for earlier detection and treatment. As new clinical entities originating from ICIs continuously emerge, such case series or case reports, these could prove as an invaluable tool for everyday clinical practice. 

## Figures and Tables

**Figure 1 jcm-11-04786-f001:**
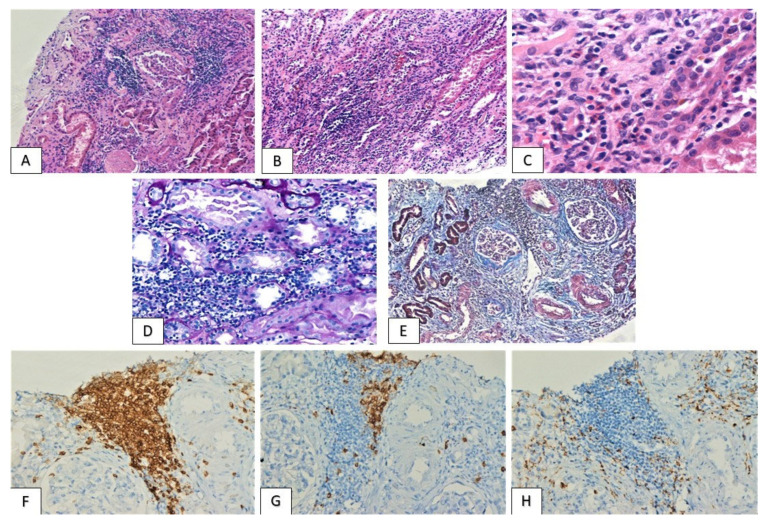
Tubulointerstitial nephritis: focal ((**A**): HE ×100) and diffuse ((**B**): HE ×100) interstitial inflammation with tubulitis lesions ((**D**): PAS ×200) and areas of interstitial fibrosis and tubular atrophy ((**E**): Masson Trichrome ×50). Infiltrates are dominated by T-cells ((**F**): CD3 ×200), while B-cells are less prominent ((**G**): CD20 ×200). Occasional eosinophils ((**C**): HE ×400) and scattered macrophages are also detected ((**H**): PGM-1 ×200).

**Figure 2 jcm-11-04786-f002:**
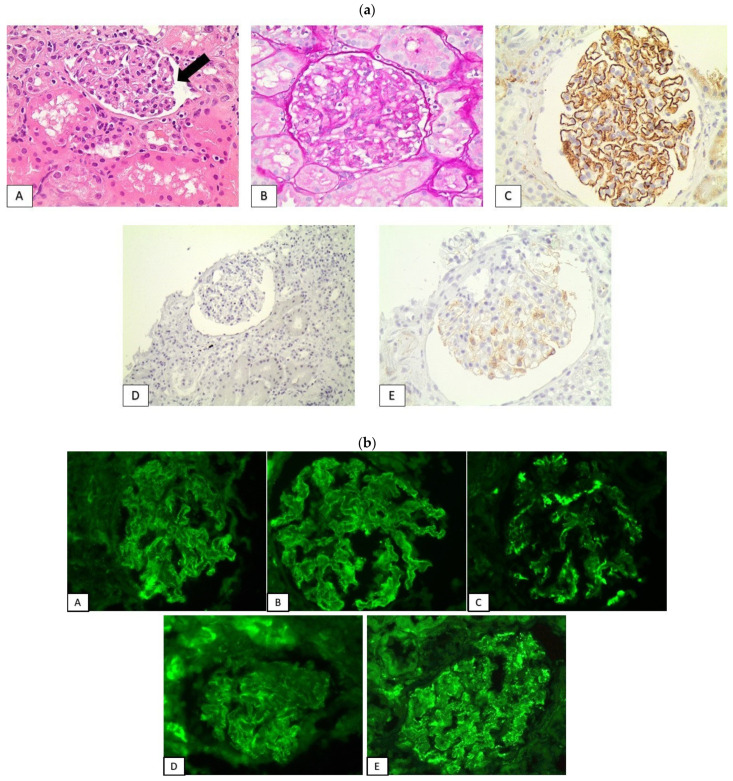
Secondary “lupus-like” membranous glomerulopathy. (**a**) Glomeruli showcased the diffuse thickening of glomerular basement membranes ((**A**): H&E ×200, (**B**): PAS ×200), while immunohistochemistry revealed the diffuse granular deposition of C4d along glomerular basement membranes ((**C**): C4d ×400). Immunohistochemical evaluation for primary membranous glomerulopathy markers IgG4 and PLA2r was negative ((**D**): IgG4 ×200, (**E**): PLA2r ×400). (**b**) A “full house” pattern, with granular IgG, IgM, IgA, C3 and C1q deposits along glomerular capillaries basement membranes was observed in immunofluorescence ((**A**): IgG ×400, (**B**): IgA ×400, (**C**): IgM ×400, (**D**): C3 ×400, (**E**): C1q ×400).

**Figure 3 jcm-11-04786-f003:**
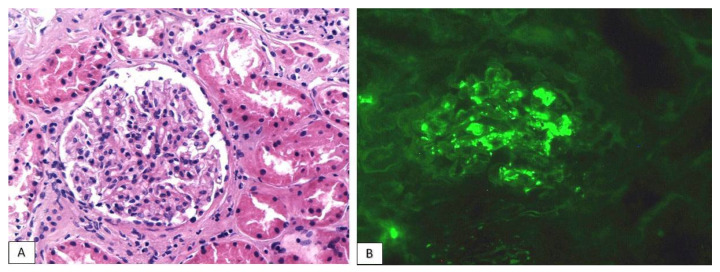
IgA glomerulonephritis with mild mesangial hypercellularity and mesangial matrix expansion ((**A**): HE ×200). Immunofluorescence shows strong mesangial deposition of IgA immunoglobulin ((**B**): IgA ×200).

**Figure 4 jcm-11-04786-f004:**
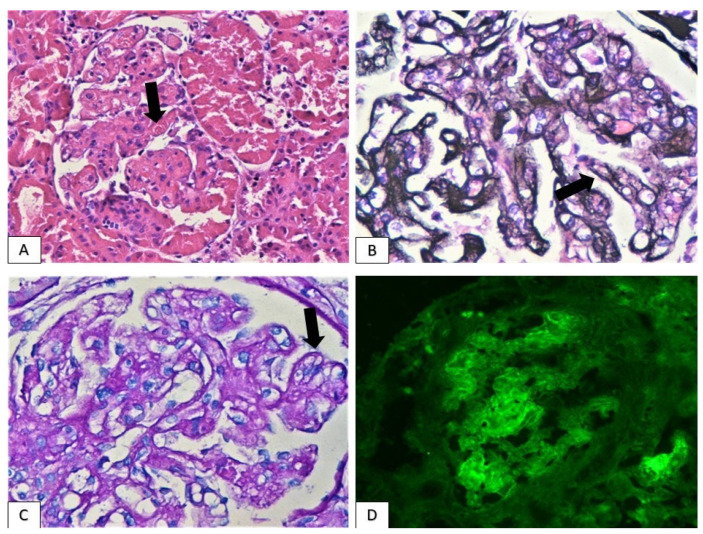
Glomerulus with intraglomerular fibrin thrombus (arrow, (**A**): H&E ×400) and segmental double contours of glomerular basement membranes (arrows, (**B**): silver ×400 and (**C**): pas ×400). ((**D**): immunofluorescence showing IgG deposits along glomerular basement membranes and mesangium, ×400).

**Figure 5 jcm-11-04786-f005:**
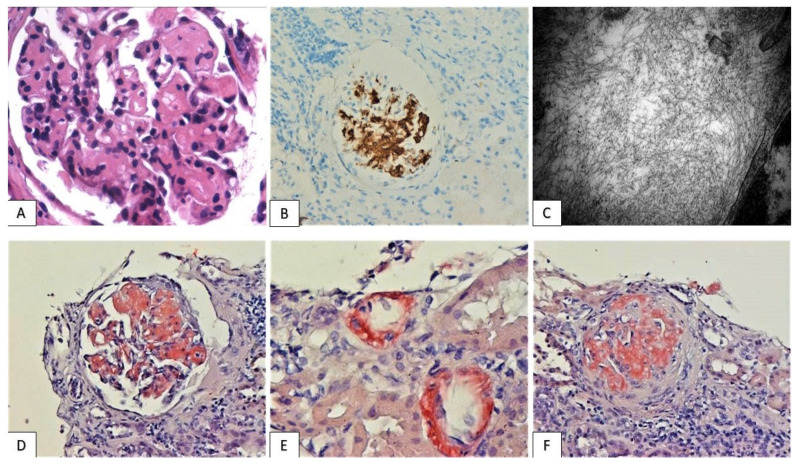
Secondary AA amyloidosis: Glomeruli displayed expansion of the mesangial area, which was occupied by acellular eosinophilic aggregates ((**A**): H&E ×400). These amorphous aggregates stained positively for Congo-Red ((**D**,**F**): ×200) and demonstrated strong diffuse immunoreactivity for amyloid A protein (AA) ((**B**): Amyloid A protein ×200). Similar Congo-Red reactive eosinophilic deposits were detected on arterioles and interlobular arteries ((**E**): ×400). ((**C**)): electron microscopy shows randomly arranged, non-branching fibrils (×44,000).

**Figure 6 jcm-11-04786-f006:**
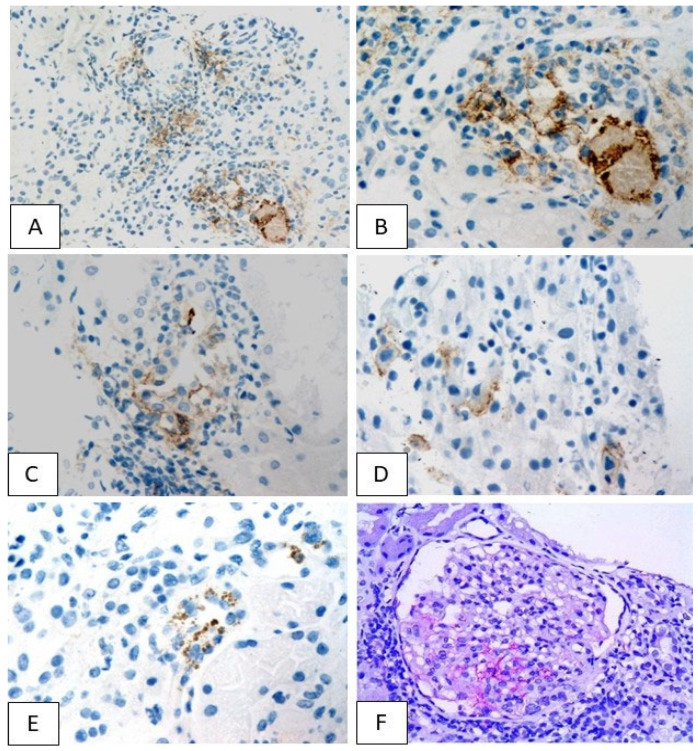
Tubular PD-L1 staining was either membranous ((**A**): 200×, (**B**–**D**): 400×) or cytoplasmic ((**E**): 400×). Glomerular PD-L1 expression was detected in a few cases ((**F**): 400×).

**Table 1 jcm-11-04786-t001:** Patients’ demographics and clinical characteristics. Yes: +, No: -.

Patients	Gender	Age	Malignancy	Immunotherapeutic Agents	Latent Period (Months)	Other Adverse Effects	PPIs
1	Male	73	bladder cancer	atezolizumab	12	-	-
2	Male	69	bladder cancer	nivolumab	12	-	-
3	Female	64	lung cancer	pembrolizumab	24	-	-
4	Male	73	lung cancer	nivolumab/ipilimumab	3	hepatitis	+
5	Male	57	renal cell carcinoma	pembrolizumab	14	-	-
6	Male	73	Merkel cell tumor	avelumab	3	-	+
7	Female	57	squamous cell cervical carcinoma	pembrolizumab	16	-	-
8	Male	66	melanoma	ipilimumab	3	rheumatic polymyalgia	-
9	Male	58	lung cancer	pembrolizumab	19	pneumonitis	+
10	Female	69	lung cancer	pembrolizumab	23	dermatitis, colitis	+
11	Male	68	lung cancer	nivolumab	10	-	-
12	Female	70	lung cancer	pembrolizumab	28	rheumatoid arthritis, thyroiditis, pancreatitis	-

**Table 2 jcm-11-04786-t002:** This table summarizes the main clinical manifestations of renal toxicity in our patients and the prominent histological patterns encountered in their biopsies. Yes: +, No: -.

Patients	Renal Failure	Creatinine (mg/dL)	Hematuria	Proteinuria	Urine Protein (g/24 h)	Nephrotic Syndrome	Tubulointerstitial Nephritis	Glomerulopathy
1	+	2	+	-	-	-	moderate	IgA glomerulonephritis
2	+	3.9	-	-	-	-	severe	-
3	+	2	-	+	0.3	-	severe	-
4	+	2.8	-	-	-	-	moderate	-
5	+	2.2	+	-	-	-	severe	-
6	+	2.5	+	-	-	-	mild	-
7	+	2.2	-	+	3.2	-	mild	Membranoproliferative glomerulonephritis/ thrombotic microangiopathy
8	+	1.8	-	+	0.254	-	mild	-
9	+	4.26	-	+	0.8	-	mild	N/A (renal medulla)
10	-	normal	-	+	16	+	mild	“lupus-like” secondary membranous glomerulopathy
11	-	normal	-	+	11	+	mild	“lupus-like” secondary membranous glomerulopathy
12	-	normal	-	+	10	+	mild	Secondary AA amyloidosis

**Table 3 jcm-11-04786-t003:** This table summarizes the chronicity index and the immunofluorescent findings of our patients’ biopsies.

Patients	Global Glomerulosclerosis (%)	IF/TA	Arterial Fibrous Intimal Thickening	IgG	IgA	IgM	C3	C1q
1	41.66	moderate	moderate	Glomeruli (1+)	Glomeruli (3+)	-	Glomeruli (1+)	-
2	12.5	severe	moderate	-	-	-	-	-
3	11.7	mild	moderate	-	-	-	-	-
4	29.4	moderate	moderate	-	-	Glomeruli (1+)	-	-
5	NA (renal medulla)	NA (renal medulla)	No arteries	-	-	-	-	-
6	33.3	mild	severe	-	-	Glomeruli (1+)	-	-
7	0	mild		Glomeruli (2+)	Glomeruli (1+)	Glomeruli (2+)	Glomeruli (2+)	-
8	22.22	mild	severe	-	-	-	-	-
9	NA (renal medulla)	NA (renal medulla)	No arteries	-	-	-	-	-
10	18.18	moderate	severe	Glomeruli (2+)	Glomeruli (2+)	Glomeruli (2+)	Glomeruli (2+)	Glomeruli (2+)
11	25	moderate	moderate	Glomeruli (3+)	Glomeruli (1+)	Glomeruli (1+)	Glomeruli (3+)	Glomeruli (2+)
12	35.48	mild	moderate	-	Glomeruli (1+)	Glomeruli (1+)	Glomeruli (2+)	Glomeruli (2+)

NA: Not Available.

**Table 4 jcm-11-04786-t004:** The table summarizes PD-L1 staining patterns in all twelve patients. Positive stain: +, negative stain: -.

Patients	Tubular Epithelium		Glomeruli (Intensity)
	Cytoplasmic (Intensity)	Membranous (Intensity)	
1	+(mild)	-	-
2	+(moderate)	-	-
3	+(mild)	-	-
4	+(severe)	+(severe)	-
5	+(mild)	-	NA
6	+(severe)	+(severe)	-
7	+(moderate)	+(moderate)	-
8	-	-	-
9	+(moderate)	-	NA
10	-	-	+(moderate)
11	+(moderate)	-	+(mild)
12	+(mild)	-	+(mild)

**Table 5 jcm-11-04786-t005:** The table summarizes the therapeutic approach of patients’ immune-mediated kidney toxicity and their response to treatment. Yes: +, No: -.

Patients	Immunotherapy Withdrawal	Immunosuppression	Kidney Response	Hemodialysis
1	NA	NA	NA	NA
2	+	prednisolone	No remission	+
3	+	prednisolone	remission	-
4	+	prednisolone	remission	-
5	+	prednisolone	remission	-
6	+	prednisolone	remission	-
7	NA	NA	NA	NA
8	NA	NA	NA	NA
9	+	prednisolone	remission	-
10	+	prednisolone	remission	-
11	NA	NA	NA	NA
12	+	prednisolone and colchicine	remission	-

## Data Availability

The data presented in this study are available on request from the corresponding author.

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
