# Peer review of "Immune Checkpoint Inhibitors’ Associated Renal Toxicity: A Series of 12 Cases"

_jcm, 2022, doi:10.3390/jcm11164786_

Round 1

Reviewer 1 Report

The paper of Palamaris represents another case series on the hot topic of cpi renal toxicity. The originality of the data is not great except for the immunofluorescence data but the cases are well described and the paper could increase the knowledge in this field with some enrichment. These are my suggestions:

Cassol et al. proposed that the evaluation of PD-L1 expression by immunohistochemistry could be a marker to differentiate AIN related to cpi from AIN of other etiology. Successively this was confirmed a with some limitations by Hakroush . Thus, it would be important to evaluate PD-L1 expression in this series also relating Pd-L1 distribution to the different compartments injured.

Furthermore, it would be useful to describe (also semi-quantitatively) the intensity of inflammatory infiltrate and try a correlation to the response to the therapy as Di Giacomo  attempted in a recent paper. Finally, the discussion could reserve some comment on the therapeutic approach in light of  new strategy recently experimented  (infliximab).

Finally, at line 347 the statement- the first case of AA amyloidosis  where the specific systemic autoimmune background that provided the initial trigger for amyloid deposition has been identified – is not clear and should be reformulated because the pathogenetic link between autoimmune disorders and the cpi toxicity is well known and the amyloid deposition has not any proved pathogenic relationship with the cpi toxicity.    

Author Response

We thank the reviewer for the constructive comments. Please find below our response to your comments

  1. “Cassol et al. proposed that the evaluation of PD-L1 expression by immunohistochemistry could be a marker to differentiate AIN related to cpi from AIN of other etiology. Successively this was confirmed a with some limitations by Hakroush . Thus, it would be important to evaluate PD-L1 expression in this series also relating Pd-L1 distribution to the different compartments injured.”

We have evaluated PD-L1 expression, as described in the newly added section 3.3, as well as in table 4 and figure 6.

  1. “Furthermore, it would be useful to describe (also semi-quantitatively) the intensity of inflammatory infiltrate and try a correlation to the response to the therapy as Di Giacomo attempted in a recent paper.”

The intensity of inflammatory infiltrate is described in table 2. In our cohort, we did not observe a correlation between the intensity of inflammatory infiltrate and response to treatment, as all patients but one responded to treatment.  However, we noticed that the patient with progressive renal disease demonstrated the highest level of chronicity, as referred in section 3.4 and in the Discussion. Di Giacomo’s paper was added to references (ref. 36).

  1. “Finally, the discussion could reserve some comment on the therapeutic approach in light of new strategy recently experimented (infliximab).”

We have added a paragraph referring to the common and novel therapeutic approaches to immune-related kidney adverse effects, according to the reviewer’s suggestion.

  1. “ Finally, at line 347 the statement- the first case of AA amyloidosis where the specific systemic autoimmune background that provided the initial trigger for amyloid deposition has been identified – is not clear and should be reformulated because the pathogenetic link between autoimmune disorders and the cpi toxicity is well known and the amyloid deposition has not any proved pathogenic relationship with the cpi toxicity.”

While rheumatoid arthritis (RA) is a well-known cause of secondary amyloidosis, this is the first time to our knowledge, that a case of AA amyloidosis as a consequence of RA due to pembrolizumab, is described.

The sentence in the above phrase (line 393 in the revised ms) has been restructured according to the reviewer’s suggestions.

Reviewer 2 Report

Dr. Palamaris and colleagues present a series of 12 patients with immune checkpoint inhibitor induced nephritis. I have a few comments, questions, and concerns:

1. Table 1 - it is not necessary to label each drug at each mention (e.g. anti-PD-1) - either a footnote or labeling at first mention should suffice. Also avelumab and atezolizumab are both anti-PD-L1 drugs, not anti-PD-1.

2. Several places in the manuscript, PD-1 and PD-L1 blockers are either mislabeled or incorrectly mentioned together. 

3. When discussing case vignettes, please list patient number so that readers can match the cases with the tables

4. Did any patients require 2nd line immunosuppression (e.g. mycophenolate)?

5. The authors do not make any case why their series is novel/different/worth publishing compared with prior series - they should at least include a few sentences. 

6. The authors do not cite many of the main large case series - please cite PMID: 34625513 and PMID: 31896654 and PMID: 31672794

Author Response

We thank the reviewer for the constructive comments. Please find below our response to your comments

  1. “Table 1 - it is not necessary to label each drug at each mention (e.g. anti-PD-1) - either a footnote or labeling at first mention should suffice. Also avelumab and atezolizumab are both anti-PD-L1 drugs, not anti-PD-1.”

According to the reviewer's suggestions, we have removed the labeling of each drug from Table 1. 

  1. “ Several places in the manuscript, PD-1 and PD-L1 blockers are either mislabeled or incorrectly mentioned together.”

We have corrected the mentioning/labeling of PD-1, PD-L1 inhibitors in the text.

  1. “When discussing case vignettes, please list patient number so that readers can match the cases with the tables.”

We have added in the text the patients’ numbers according to Table 1.

  1. Did any patients require 2nd line immunosuppression (e.g. mycophenolate)?”

None of our patients received second line immunosuppressive treatment.

  1. “The authors do not make any case why their series is novel/different/worth publishing compared with prior series - they should at least include a few sentences.”

We have added a few sentences in “Conclusions” regarding some findings reported for the first time in our study and pointed out the need for studies like this, in order to better understand and characterize the wide range of immunotherapy related renal toxicity

  1. “The authors do not cite many of the main large case series - please cite PMID: 34625513 and PMID: 31896654 and PMID: 31672794”

We have added two of the suggested references in the discussion section (references 23, and 24), according to the reviewer’s recommendation. The reference with PMID: 31896654 seems irrelevant to the topic and so it was not included.

Round 2

Reviewer 1 Report

The quality of the paper has improved and now it represent a real

contribution in the field of the cpi associated renal toxicity